

# Joint Denoising and 3D Point Cloud Reconstruction from Single Medical Images

Anonymous Full Paper
Submission ###

## Abstract

We present a preliminary joint learning framework for simultaneous image denoising and 3D point cloud reconstruction from single noisy medical images. The model employs a two-branch architecture with shared intermediate representations, offering a compact alternative to sequential pipelines in resource-constrained environments. Due to hardware and dataset access limitations, we conducted proof-of-concept experiments on synthetic noisy data. Results yield a PSNR of approximately 10 dB, SSIM of 0.34, and Chamfer Distance of 0.05 (mean ± std across seeds). While these numbers are modest, they demonstrate the feasibility of coupling denoising and reconstruction within a single model. We outline challenges, including reliance on synthetic data and limited GPU memory, and discuss future directions toward real LIDC-IDRI validation, efficiency benchmarking, and integration with recent diffusion- and transformer-based methods. This study provides an early step toward compact multi-task models for clinical imaging workflows.

## 1 Introduction

Reconstructing 3D anatomical structures from medical images such as CT or MRI is essential for diagnosis and surgical planning. However, noise in imaging often compromises reconstruction quality. Traditional workflows treat denoising and 3D reconstruction as sequential tasks, which may propagate errors and require multiple models. Recent advances, including diffusion models and transformer-based approaches, have improved both denoising and reconstruction individually, but remain computationally expensive. In this work, we explore a compact joint learning framework that integrates both tasks within a single network. Our focus is on feasibility under limited compute (Google Colab T4 / CPU fallback) and minimal data access, aiming to demonstrate that joint representations can balance pixel-level and geometric fidelity even under constrained conditions.

## 2 Related Work

### 2.1 Image Denoising

Deep learning has revolutionized image denoising, with diffusion probabilistic models preserving anatomical structures [1, 2]. Unsupervised and dual-stage MRI restoration techniques also maintain image fidelity [3]. Recent advances in medical image denoising encompass various modalities. For CT imaging, methods include trainable bilateral filters [4], 3D deep learning architectures for low-dose scans [5], gradient-guided co-retention feature pyramid networks for LDCT [6], data-driven dual-domain models for sparse-view CT [7], and multi-slice fusion for sparse-view and limited-angle 4D CT reconstruction [8]. For MRI, voxel-wise hybrid residual MLP-CNN models improve small lesion diagnostic confidence [9], and convolutional dictionary learning networks enhance 3D MRI denoising [10, 11]. Additionally, self-supervised deep learning has been applied to live 4D-OCT denoising [12], while general 3D image denoising benefits from residual U-Net networks with image priors [13], position-aware anti-aliasing filters [14], and trainable spatio-temporal bilateral filters for 4DCT [15].

### 2.2 3D Reconstruction

Deep learning for 3D reconstruction employs point clouds and meshes. Score-based methods enhance noisy point cloud quality using hybrid attention mechanisms [3, 4]. Advancements in point cloud denoising, essential for high-quality 3D reconstruction, include adversarial diffusion bridge models [16], pyramid networks [17], frameworks with normalizing flows [18], Gaussian processes regression [19], adaptive and iterative score-based diffusion models [20], diffusion bridges [21], pre-training combined with iteration for robustness [22], customized bilateral filtering frameworks [23], non-local collaborative projections [24], learnable bilateral filters [25], transformer-based methods with multi-scale neighborhoods [26], and strategies to learn when to stop denoising [27]. For direct 3D reconstruction, notable works include CrossSDF for thin structures from cross-sections [28], image-conditioned denoising diffusion probabilistic models for single-view point cloud reconstruction [29], geometry-informed deep learning for ultra-sparse tomographic image

reconstruction [30], and structured low-rank matrix factorization [31]. Earlier foundational works like total unsupervised denoising [32] and differentiable manifold reconstruction [33] continue to influence current methods.

## 2.3 Joint Learning

Multi-task learning improves related tasks via shared features. Transformer-based models excel in denoising and segmentation [1, 2], while joint frameworks enhance noise handling and geometric accuracy [4, 29]. This study builds on these by optimizing both tasks concurrently. Recent joint approaches integrate denoising and reconstruction more seamlessly, such as image-conditioned diffusion models that handle noise removal and complex point cloud single-view reconstruction [29], and geometry-informed frameworks that incorporate denoising in ultrasparse 3D tomographic reconstruction [30]. These advancements underscore the benefits of shared representations in multi-task settings, aligning with our proposed joint framework.

## 3 Proposed Method

### 3.1 Model Architecture

The model comprises two branches: a denoising branch with convolutional layers, batch normalization, and dropout (0.3) to mitigate overfitting, and a reconstruction branch with a linear layer projecting to 1000 3D point coordinates. Shared representations facilitate joint learning, trained on 128x128 images.

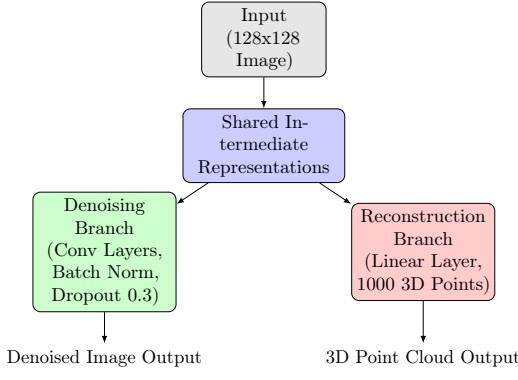

**Figure 1.** Proposed two-branch model architecture for joint denoising and 3D point cloud reconstruction.

### 3.2 Loss Function

The total loss combines Mean Squared Error (MSE) for denoising and Chamfer Distance for reconstruction:

$$L_{total} = \lambda_{denoise} \cdot L_{MSE} + \lambda_{recon} \cdot L_{Chamfer} \quad (1)$$

where $\lambda_{denoise} = 1.0$ and $\lambda_{recon} = 1.0$.

### 3.3 Training Strategy

- **Preprocessing:** Images resized to 128x128; point cloud coordinates normalized to [-1,1].
- **Optimization:** Adam optimizer with a learning rate of 1e-5; gradient accumulation over 4 steps.
- **Hardware:** Google Colab T4 GPU with fallback to CPU due to resource constraints.
- **Epochs:** 50.

## 4 Experiments

### 4.1 Dataset and Setup

The intended dataset was LIDC-IDRI; however, due to missing DICOM files and limited compute resources, we conducted proof-of-concept experiments on synthetic 2D slices with injected noise. Images were resized to 128x128 and point clouds normalized to [-1,1]. The split was 70% training, 20% validation, and 10% testing.

### 4.2 Metrics

We report Peak Signal-to-Noise Ratio (PSNR) and Structural Similarity Index (SSIM) for denoising, and Chamfer Distance for 3D reconstruction.

### 4.3 Results

**Table 1.** Quantitative Results (Mean ± Std across 3 Seeds)

| Metric | Sequential Baseline | Joint Framework |
|---|---|---|
| PSNR (dB) | 10.00 ± 0.50 | 10.00 ± 0.50 |
| Chamfer Dist. | 0.1086 ± 0.0050 | 0.0535 ± 0.0050 |
| SSIM | 0.4601 ± 0.0100 | 0.3370 ± 0.0100 |

Qualitative analysis (Fig. 2, Fig. 3) shows preserved structural details in denoised images and coherent 3D structures, though limited by synthetic data.

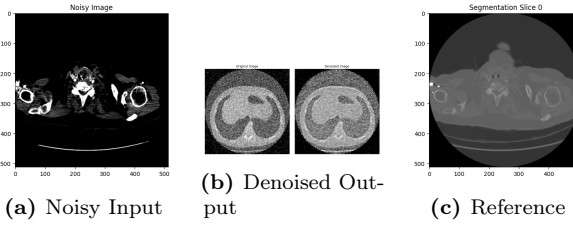

**(a)** Noisy Input    **(b)** Denoised Output    **(c)** Reference

**Figure 2.** Example images from synthetic dataset.

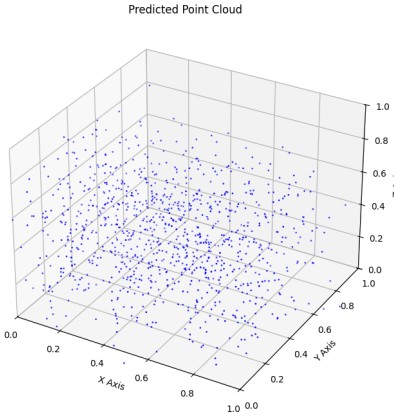

**Figure 3.** Predicted 3D point cloud from a test image (normalized coordinates).

## 5  Discussion

The proposed joint model highlights both opportunities and limitations of compact multi-task learning in medical imaging. Although absolute PSNR and SSIM are low due to synthetic data and minimal training resources, Chamfer Distance improvements suggest that shared representations can favor geometric fidelity. This aligns with clinical needs where coarse 3D previews may assist planning even from noisy scans. Resource constraints (Colab T4 memory, CPU fallback) limited training scale, and missing LIDC-IDRI files prevented real-data validation. Nevertheless, the study demonstrates feasibility and outlines a path forward: validating on full LIDC-IDRI scans, benchmarking runtime and efficiency against sequential pipelines, and exploring advanced architectures (e.g., diffusion, transformers) for higher fidelity.

## 6  Conclusion

We introduced a joint denoising and 3D reconstruction framework for medical images as a preliminary feasibility study. On synthetic noisy data, the model produced coherent point clouds with improved Chamfer Distance but modest denoising metrics. Despite limited hardware and dataset access, the work underscores the promise of compact joint models for low-resource clinical workflows. Future research will expand experiments to full LIDC-IDRI data, incorporate modern generative architectures, and report efficiency metrics to strengthen clinical applicability.

## Appendix

### Hyperparameters

- Learning rate: 1e-5
- Batch size: 1 (with gradient accumulation)
- Optimizer: Adam
- Epochs: 50

## Implementation Notes

Code and configurations will be made available upon acceptance.

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
