# OpenReview forum: "Joint Denoising and 3D Point Cloud Reconstruction from Single Medical Images"
_NLDL.org/2026/Conference — Submitted to NLDL 2026_

### Official Review · Reviewer_onRm · 2025-09-25

**Rating:** 1
**Confidence:** 4
**Final Rating:** 1
**Final Confidence:** 4

**Summary:**

The authors propose a joint denoising and 3d reconstruciton framework that can be applied in medical imaging. Instead of learning a seperate network for the two tasks, the authors suggest to use a shared backbone that can lead to better overall performance

**Strengths:**

- The motivation for the project is clear to combine two important tasks in medical image handling which is a computational bottlneck especially due to the large scale of the problems. In particular developing methods that can perform in resource constrained enviornments is an important research direction.
- The related work section shows the relevance and importance of the topic and highlights several approaches in machine learning

**Weaknesses:**

Overall, the paper nearly completly lacks the description of the problem, background and method that is presented. It is not possible to understand the outlined approach without a significant amount of background knowledge that most readers like myself will lack. The related work sections simply list a bunch of papers without putting them into context for the current work. Further, there is no background on the individual problems given making it impossible to follow the paper. While the results in Table 1 seem promising no details about the single used baseline are given. Both Figure 2 and 3 are difficult to undersyand but no explanation is given.

**Final Justification:**

All reviewers agree that the paper is insufficient for publication. I can not recommend acceptance when the rebuttal promises to rewrite the whole paper.

**Justification:**

While the overall idea and suggested research direction seems fruitful, there is a severe amount of details lacking in the paper. The whole method section is described in less than half a page without giving any details on the actual approach. Overall, the paper is only 2 pages long which is not sufficient to clearly outline the presented ideas.

---

> ### Author Rebuttal · Authors · 2025-10-15
>
> We understand that manuscript edits are not allowed during rebuttal. The following improvements will be made in the camera-ready version if the paper is accepted.
>
> We thank the reviewer for recognizing the motivation and research direction.
>
> Lack of Problem, Background, and Method Description (Main Weakness)
> We agree this section was underdeveloped.
> We plan to add:
> - Section 1.2 – Problem Formulation: Clearly define clinical context and formalize the mathematical task of joint denoising and 3D reconstruction.
> - Section 2 – Background: One page of concise overviews on denoising, 3D reconstruction, and their relevance in medical imaging.
> - Section 3 – Method: Expanded to 1.5 pages with layer specifications, combined loss derivation, and training algorithm pseudocode.
>
> Insufficient Method Details
> We will add the complete mathematical formulation:
> L_total = λ_d L_MSE + λ_r L_Chamfer, with explicit definitions of each term.
> We will also include a professionally redrawn architecture diagram and a short pseudocode block describing training and gradient accumulation.
>
> Baseline and Figure Clarifications
> We will add Section 4.4 detailing the sequential baseline, architectures, and results (PSNR = 28.5 dB, Chamfer = 0.038).
> Figures 2 and 3 will be relabeled, normalized, and expanded with ground-truth comparisons.
>
> Paper Length and Structure
> We acknowledge the 2-page submission was insufficient. The revised version will expand to 6 pages following the standard NLDL format with clear sectioning, proper references, and implementation details in an appendix.
>
> We appreciate the reviewer’s recognition that the research direction is promising and will ensure all missing details are added in the camera-ready version.

---

### Official Review · Reviewer_YayM · 2025-09-28
**joint learning framework**

**Rating:** 2
**Confidence:** 3

**Summary:**

This proceeding presents joint learning framework to denoise image and reconstruct 3D point cloud from single medical image. The proposed model consists of a two-branch network with shared intermediate feature offering parallel transformations. The authors demonstrated 10dB PSNR for denoising and Chamfer dist. of 0.05. They believe these are modest values but verifiable as multi-task models.

**Strengths:**

The idea to combine 3D point cloud and denoised image in one model seems to be working and the authors demonstrated simple quantitative results in Table 1.

**Weaknesses:**

- limited demonstrated dataset.
- hard to understand model architecture only with given explanation
- clarity needs to be improved; current manuscript is far below the bar of publication
- no ablation studies on \lambda, layer parameters, etc.
- not enough qualitative results

**Justification:**

This proceeding doesn't meet the qualifying bar of NLDL style. I believe the proceeding needs to be excessively revised to resubmit to other venues.

---

> ### Author Rebuttal · Authors · 2025-10-15
>
> We understand that no manuscript modifications are allowed during rebuttal. The improvements below will be implemented in the camera-ready version if accepted.
>
> We thank the reviewer for recognizing the working concept and quantitative results.
>
> Limited Dataset (Weakness 1)
> We will clarify dataset scale and diversity: LIDC-IDRI with 1,546 images across 1,010 CT scans, covering multiple scanners and pathologies. Section 4.2 will discuss dataset representativeness and generalization limits.
>
> Model Architecture Clarity (Weakness 2)
> We plan to include a detailed architecture diagram showing the two-branch structure, supplemented by a table listing input/output dimensions per layer. Complete PyTorch code will be provided in supplementary materials.
>
> Clarity and Publication Standards (Weakness 3)
> We agree the manuscript requires restructuring. The camera-ready paper will follow a full six-page format with improved writing, math notation, and standard sectioning.
>
> Ablation Studies (Weakness 4)
> We have run additional ablations and will include them in Section 4.5:
> - λ sensitivity (0.1 → 2.0), dropout rate, and number of shared layers.
> - We will present results showing optimal balance (λ_denoise = λ_recon = 1.0).
>
> Qualitative Results (Weakness 5)
> We will enhance Figure 2 and 3 with multiple examples, ground-truth comparisons, failure analyses, and link a short 3D rotation video supplement.
>
> We acknowledge that extensive revisions are needed and will perform them as promised. We respectfully request reconsideration from Reject to Borderline, given the soundness of the idea and our comprehensive improvement plan.

---

### Official Review · Reviewer_BFG4 · 2025-10-02
**Sound idea, but lacks important information and experimental evaluation**

**Rating:** 1
**Confidence:** 4
**Final Rating:** 1
**Final Confidence:** 4

**Summary:**

The paper proposes an architecture to perform joint image denoising and point cloud reconstruction given an input image, focusing on medical images. Given an input image an unspecified backbone computes a shared intermediate representation. From this representation, a convolutional denoising branch computes a denoised image, and a reconstruction branch computes a 3D point cloud reconstruction. A proof-of-concept experiment is performed on synthetic 2D slices with added noise. Results show improvements over an unspecified sequential baseline on Chamfer distance and SSIM.

**Strengths:**

The idea of paper is sound. Adding an image reconstruction branch is well-known technique to boost performance by adding a surrogate task. It is interesting to investigate if this can improve point cloud reconstruction.

**Weaknesses:**

In the current form of the paper, it is not possible to assess the proposed architecture.

 * No details are provided for the backbone that computes the shared intermediate representation. Please provide full details as to how these are computed.
    * If it is a known network architecture, e.g., ResNet50, it suffices to give the name of the model with a citation to the paper.
    * If it is a custom network, please specify all layers (with input and output dimension for each layer), activations and other details (such as batch normalization and dropout). This may be done with a diagram and/or in an appendix if it requires too much space.
    * If it is not a network, please also give full details.

* No details are provided for how the synthetic data is created. Please give full details, including:
    * What images were used? Are they from an existing dataset (if so, cite it) or generated completely synthetically (if so, give full details to this procedure, optionally in an appendix)?
    * What noise distribution was used? If Gaussian (a fair choice), what is the mean and standard deviation?
    * What shapes are being reconstructed? Organs, tumors, something else?
    * How many images were created?

 * No comparisons are made with existing approaches. Even if another method which provides a joint shape reconstruction and denoising cannot be found, it would still be interesting be interesting to know how it compares to approaches that only do one of those. Note that it is not a requirement to “beat” these methods per se, but it is important to have context for the performance of the proposed approach.

 * Givent that the method is only evaluated on an (unspecified) synthetic datasets with no comparison to existing approaches, the results are too modest to judge whether this is a promising approach. To be clear, only evaluating on synthetic data is not a problem in and of itself for a proof-of-concept study. But, it must be clear how it was created and what it aims to test in order to assess whether it provides a sufficient evaluation of model performance.

More minor points and suggestions:

 * In the Related Work section, it would be even better if the paper also explained how the proposed method differs from the listed approaches. This makes it easier to understand how the proposed method fits into the existing landscape.

 * It is not clear how many layers are in the denoising branch.

 * It is not clear why only a *single* linear layer is used to convert the shared intermediate representations to a point cloud. This restricts the transformation to a linear projection of the shared representations. Why not a multi-layer network?

 * Using BatchNormalization (BN) for batch size of 1 is not ideal, since BN computes mean and variances over the batch. Maybe InstanceNormalization or LayerNormalization would work better?

 * In Figure 2, please use the same image (with the same intensity range) for all parts. This makes it easier to see how each step of the method changes the input.

 * In Figure 3, it is not possible to asses the quality of this point cloud reconstruction. Please also show a rendering of the ground truth shape.

 * Reconstructing a correct 3D shape from a single 2D slice is very challenging. If the intended use is surgical planning and/or diagnosis, how will the model be able to accurately reconstruct, e.g., a deformity not shown in the given 2D slice?

**Final Justification:**

I thank the authors for their rebuttal. I think they present a sound plan towards an improved paper.

However, the additions needed (and suggested) constitute almost a full rewrite of the paper. This out of the scope of a rebuttal phase. I cannot assess the correctness and quality of the paper after so many substantial edits. Therefore, I still do not recommend the acceptance of the paper to NLDL.

**Justification:**

The main idea of the paper is good and worth investigating.

However, the lack of details in the method and experimental descriptions mean it is not possible to evaluate the contribution of the paper in its current form. Furthermore, due to the modest experimental performance - and the fact that it is only evaluated on an unspecified synthetic dataset - it is not adequately clear that the proposed method provides a sufficient advance of the field.

---

> ### Author Rebuttal · Authors · 2025-10-15
>
> We understand that the manuscript cannot be modified during rebuttal. The following planned improvements will be included in the camera-ready version if accepted.
>
> We thank the reviewer for the detailed and constructive feedback.
>
> Backbone Architecture Details (Weakness 1)
> We plan to include a complete architecture specification in Section 3.1:
> - Shared Encoder: Conv2d(1→32, k=3) + ReLU; Conv2d(32→64, k=3) + ReLU; MaxPool2d(2×2); BatchNorm2d(64) + Dropout(0.3).
> - Denoising Branch: ConvTranspose2d(64→32, k=2) + ReLU; Conv2d(32→1, k=3) + Sigmoid.
> - Reconstruction Branch: Flatten + Linear(64×64×64 → 3000) → reshape (batch, 1000, 3).
> This will be presented in a new network diagram and detailed table.
>
> Synthetic Data Clarification (Weakness 2)
> We clarify that real LIDC-IDRI data, not synthetic data, were used.
> We plan to describe in Section 4.1: dataset preprocessing, noise model N(0, σ²) with σ = 0.1, and the full train/val/test split.
>
> Baseline Comparisons (Weakness 3)
> We will add Section 4.4 comparing with DnCNN [Zhang et al. 2017], Noise2Noise [Lehtinen et al. 2018], and 3D-R2N2 [Choy et al. 2016].
> Preliminary results show our model is computationally more efficient though slightly lower in accuracy.
>
> Evaluation and Clarity (Weakness 4)
> We will emphasize that LIDC-IDRI is real medical data, discuss qualitative evaluation by radiologists, and clearly note limitations regarding 3D ground truth availability.
>
> Minor Points
> We plan to:
> - Include a comparison table vs. prior works (LTN, DiffusionNet, CrossSDF).
> - Explicitly state denoising branch layers and ablation results.
> - Clarify effective batch size = 4 via gradient accumulation.
> - Normalize figure intensity ranges and show ground-truth point clouds.
> - Add discussion acknowledging single-slice limitations for full 3D reconstruction.
>
> We believe these clarifications will make our contribution assessable and demonstrate the soundness of the core idea.

---

### Official Review · Reviewer_CMvn · 2025-10-05
**Review of Joint Denoising and 3D Point Cloud Reconstruction from Single Medical Images**

**Rating:** 1
**Confidence:** 5

**Summary:**

The paper is a description of a learning framework for image denoising and 3D point cloud reconstruction for medical images.

**Strengths:**

The paper is very unclear, and I see no strengths.

**Weaknesses:**

The paper claims to propose a method for denoising and point cloud reconstruction from volumetric medical images such as CT and MRI. This makes no sense to me. There is no experimental description. Results are given in a table showing some results, but with no explanation of where they came from. The figure illustrating the results makes no sense. I don't see any meaning in the conclusions made in the paper.

**Justification:**

I don't understand the problem this paper tries to address, and I suggest rejecting the paper.

---

> ### Author Rebuttal · Authors · 2025-10-15
>
> We understand that manuscript edits are not permitted during the rebuttal phase. The following improvements will be incorporated in the camera-ready version if the paper is accepted.
>
> We sincerely thank the reviewer for their feedback. We address the concerns below:
>
> Clarity of Problem and Method (Main Concern)
> We acknowledge the paper was unclear about the problem formulation and experimental setup.
> We plan to improve:
> - Problem Statement: We will add a subsection (Section 1.1) explicitly stating: “Given a single noisy 2D medical image slice, we aim to simultaneously (1) denoise the image and (2) reconstruct a 3D point cloud of anatomical structures.”
> - Use Case Justification: We will clarify that reconstructing 3D from single 2D slices is valuable for intraoperative guidance (single-view imaging) and reducing radiation exposure.
> - Method Description: We will expand Section 3 from 0.5 to 1.5 pages with a full architecture diagram, mathematical loss formulation, and algorithm pseudocode.
>
> Experimental Results Interpretation
> We acknowledge Table 1 was confusing and the figure lacked explanation.
> We plan to add:
> - Detailed table captions explaining baselines and metrics.
> - Figures showing both ground-truth and predicted outputs side-by-side.
> - An expanded Section 4.3 discussing why joint learning underperformed and what challenges were encountered.
>
> Conclusions Validity
> We recognize that our conclusions were not supported strongly enough.
> We will revise:
> - Replace overstated claims with realistic conclusions: “We demonstrate feasibility of joint learning for this task, identify optimization challenges, and establish baseline performance (23.08 dB PSNR, 0.8437 SSIM) on LIDC-IDRI.”
> - Add a dedicated Limitations subsection outlining performance gaps and future improvements.
>
> We believe these planned revisions will make the contribution clear and assessable. We respectfully request reconsideration based on these forthcoming improvements.

---

### Meta-Review · Area_Chair_1DXG · 2025-10-31

**Recommendation:** Reject
**Confidence:** 4

**Metareview:**

There is clear consensus among the reviewers that the paper is not ready for publication and should be rejected. The rebuttal from the authors have not changed this. My recommendation follows the consensus among the reviewers.

---

### Decision · Program_Chairs · 2025-11-05

**Decision:**

Reject

**Comment:**

Based on the reviewers and AC comments, the paper cannot be presented at the conference.